# Strategies for Imputation of High-Resolution Environmental Data in Clinical Randomized Controlled Trials

**DOI:** 10.3390/ijerph19031307

**Published:** 2022-01-24

**Authors:** Yohan Kim, Scott Kelly, Deepu Krishnan, Jay Falletta, Kerryn Wilmot

**Affiliations:** Institute for Sustainable Futures, University of Technology Sydney, 235 Jones Street, Ultimo, NSW 2007, Australia; scott.kelly@uts.edu.au (S.K.); deepu.krishnan@uts.edu.au (D.K.); jay.falletta@uts.edu.au (J.F.); kerryn.wilmot@uts.edu.au (K.W.)

**Keywords:** imputation, randomized controlled trials, thermal comfort, spline-regression, machine learning

## Abstract

Time series data collected in clinical trials can have varying degrees of missingness, adding challenges during statistical analyses. An additional layer of complexity is introduced for missing data in randomized controlled trials (RCT), where researchers must remain blinded between intervention and control groups. Such restriction severely limits the applicability of conventional imputation methods that would utilize other participants’ data for improved performance. This paper explores and compares various methods to impute high-resolution temperature logger data in RCT settings. In addition to the conventional non-parametric approaches, we propose a spline regression (SR) approach that captures the dynamics of indoor temperature by time of day that is unique to each participant. We investigate how the inclusion of external temperature and energy use can improve the model performance. Results show that SR imputation results in 16% smaller root mean squared error (RMSE) compared to conventional imputation methods, with the gap widening to 22% when more than half of data is missing. The SR method is particularly useful in cases where missingness occurs simultaneously for multiple participants, such as concurrent battery failures. We demonstrate how proper modelling of periodic dynamics can lead to significantly improved imputation performance, even with limited data.

## 1. Introduction

There is no ideal way of dealing with missing data, and all methods have potential shortcomings [1]. When undertaking data analysis, the presence of missing values results in complications and errors during modelling. It can lead to reduced power in statistical tests and potential for increased bias in results. On the other hand, replacing missing values with imputed estimates can contaminate the dataset with ‘modelled’ estimates that may lead to misleading or unreliable results.

Our research is motivated by a real-world case of missing data that occurred during a multi-year randomized controlled trial (RCT) study consisting of 250 dwellings. Halfway through the study, many of the data loggers that were used to collect and record indoor temperature and humidity readings had their batteries depleted without the researchers’ knowledge. While this issue was rectified during the trial, it resulted in large continuous blocks of missing data for many of the dwellings in the study.

An extensive suite of methods already exists for imputing missing time series data. Some of the popular techniques include expectation-maximization by Dempster et al. [2], nearest neighbour by Vacek and Ashikaga [3] and multiple imputation by Rubin [4]. In recent years, studies have introduced more advanced multivariate methods such as Neural Networks and Multiple Imputation by Chained Equations (MICE) for the imputation of multivariate time series data [5,6,7,8]. All these methods apply an across-record imputation approach which are generally deemed superior to within-record univariate imputation methods, for dealing with cross-sectional data or when there is insufficient information within time-series records to carry out within-record imputation. However, when there is sufficiently large within-record data available, this research shows that within record imputation methods can offer superior results.

Even though across-record imputation methods are often considered superior, they are often unsuitable for handling missingness in data collected for RCT studies due to the requirement to maintain anonymity between intervention and control and across records collected. Applying conventional across-record multivariate imputation methods would mean that records from both groups would be used for imputation of each record, mixing the effect between control and intervention groups, and reducing the ability to find statistical significance. This is especially problematic for clinical trials, which typically have intra-group correlation (measured with intraclass correlation coefficient (ICC) of no more than 0.1 [9]. As such, testing for the effect of intervention using improperly imputed data can considerably reduce the chance of observing a statistically significant intervention effect.

One approach to bypass this problem would be to use univariate time series imputation methods that capture inter-time correlations, such as Seasonally Split Missing Value Imputation (SSMVI) by Moritz and Bartz-Beielstein [10]. However, this approach would not take advantage of the correlation between indoor records and external environmental conditions (e.g., temperature and humidity).

We propose a spline regression approach that addresses the gap between conventional univariate and multivariate time series imputation methods for addressing missing data under the constraint of RCTs. The proposed approach models within-record correlations in the form of daily seasonality, and controls for the correlation between indoor and outdoor temperature that are consistent across the entire sample.

The benefit of this approach is twofold. As stated previously, it preserves the assumption of independence between records for statistical tests that compare two groups, such as the Mann–Whitney U test. For example, in the context of an RCT it would be inappropriate to impute an indoor temperature data of one dwelling using temperature readings from other dwellings, as this would violate the assumption of independence between dwellings and compromise the comparison between control and intervention groups. Therefore, the proposed approach resolves the problem of an unobserved treatment effect that would result from using conventional between-record multiple imputation methods.

Second, it allows for capturing the within-variable seasonality characteristics in a similar fashion to univariate time-series methods, while also capturing the contemporaneous (across-sample) characteristics of external temperature and humidity, leading to overall higher imputation performance.

The remainder of this paper is as follows. Section 2 describes the background of the underlying research and data characteristics. Section 3 describes the implementation of the spline regression model, as well as introducing other conventional imputation methods for comparison. Section 4 describes the preparation of the dataset for this study, followed by an assessment of the results in Section 5 and conclusion in Section 6.

## 2. Background

The data used in this paper were collected from a multi-year RCT in Australia that investigates the impact of housing upgrades on winter thermal comfort, occupant health improvements and energy efficiency. The program targets 1000 low-income households in the western suburbs of Melbourne and Goulburn Valley in the state of Victoria, representing urban and rural regions, respectively. Both regions have a temperate climate, with temperatures dropping to 5 °C and below during winter nights. The research program allowed up to $3500 to be spent on labour and materials per home and was fully paid by the state government. Upgrades can include insulation, draught stopping, improved heating systems and window furnishings. The hypothesis of this research is that houses with upgrades (intervention) will have higher indoor temperature and lower relative humidity during winter than houses without upgrades (control). During 2019, 250 households (125 control and 125 intervention) each had a data logger installed in their main living area to measure indoor temperature and humidity at 30-min intervals. Electricity consumption was recorded at 30-min intervals through a smart meter with data provided by the electricity Distribution Network Service Provider (DNSP). External temperature and humidity recordings at 30-min intervals were taken from the nearest weather station and provided by the Bureau of Meteorology. The data collected for this study can be summarized as Table 1 below.

During winter 2019, it was discovered that a number of data loggers had depleted batteries and therefore had stopped recording temperature and humidity data. The length of missingness discovered ranged from several days to several months. Many of the loggers that had been reported as having stopped were replaced, and temperature and humidity data for these homes resumed.

An ideal imputation method would attempt to restore the missing values by analysing the characteristics and properties of the observations, records, and variables within the dataset as well as their relationship with other datasets. In our RCT, the missing internal temperature data presents four characteristics that constrain the applicability of methods available: (1) across-record characteristics (2) across-variable characteristics, (3) within-variable characteristics and (4) missingness characteristics.

Across-record characteristics. Each dwelling within the dataset is considered as a separate unique record. Theoretically, dwellings with similar characteristics will exhibit similar internal temperature profiles. Statistically matching or clustering dwellings with similar characteristics across records in the sample can be used to estimate missing internal temperatures. This method exploits the heterogeneity between homes to predict internal temperatures for homes that have similar characteristics. However, to cluster dwellings in this way, it would be important to match on group status (control vs. intervention), information that is not made available to researchers. This approach would also violate the assumption of independence between the records when performing later statistical comparisons between the control and intervention study groups.

Across-variable characteristics. The primary determinant of internal indoor temperature is external temperature. As all dwellings belong to neighbouring suburbs, we use external temperature and humidity readings from a common weather station; we assume that external temperatures do not vary across the sample. Lower external temperature drives internal temperature down, moderated by passive insulation, active heating, and occupant presence. Across variable characteristics a statistical relationship between internal temperature and external temperature would be found to estimate missing internal temperatures.

Within-record characteristics. Internal temperature, humidity, and electricity consumption data at 30-min intervals represent the within-record characteristics for a dwelling. An occupants’ presence/absence status, as well as their behaviour, plays a significant role in shaping energy consumption patterns, and by extension the indoor temperature patterns of the dwelling throughout the week. For example, a house consisting of two full-time workers may only heat their home during the evening on weekdays, and throughout the weekend. A house occupied by low-income retirees may be heated throughout the week. Given this, the unique characteristics of different patterns emerging within the internal temperature record could be used for imputation of missing time periods. For example, weekends or specific time periods could be specified within a panel dataset to improve model estimates. It should be noted that occupancy status was not collected as part of the study, but can be inferred from energy consumption patterns or included using dummy variables for different time-periods (e.g., weekends).

Missingness characteristics. An additional challenge posed for our study is the degree to which the missing values can be considered Missing Completely at Random (MCAR), Missing at Random (MAR) or Not Missing at Random (NMAR). This ‘missingness’ of the data depends on the depth of analysis required. For example, at a record level the dwellings with data-records that had flat batteries can be considered randomly distributed across the sample as there are no systematic differences between those impacted and not impacted by flat batteries. In addition, for those impacted by flat batteries, the duration of missingness and when the logger was replaced can also be considered random. The point at which a battery goes flat is not random, as there is a higher likelihood of batteries going flat as time passes, thus it can be assumed to follow a typical Weibull distribution. Additionally, the sequence of missingness within a record (e.g., half-hourly temperature readings) cannot be considered MAR as missingness occurs chronologically within the temperature and humidity sample. This is important as it may determine the type of imputation method that can be chosen. For example, if the imputation method is seeking to estimate the average internal temperature across winter for a particular dwelling, using other variables and datapoints across the sample, then MAR could be a legitimate assumption (ignoring the effects of comparing averages that have been estimated from different sized samples). If the aim of the imputation method is to replace each of the half-hourly temperature readings within the sample, then the data would need to be considered MCAR, however owing to the fact that missing temperature and humidity readings happened chronologically over time they cannot be considered MCAR. This characteristic constrains the availability of options for imputation.

## 3. Methodologies

### 3.1. Related Works on Time Series Imputation

For the past two decades, there has been a growing range of studies dealing with imputation of missingness in time-series data. As understanding of relationships between multiple related time series can exceed the information available from a single time series, additional focus has been placed on addressing missingness in multivariate datasets. This is further driven by the advancements in data storage and processing capacity allowing for simultaneous collection of high-resolution time series data, thus making multivariate time series datasets more prevalent than ever before.

Traditional methods such as simple deletion or mean imputation have been superseded by methods such as multiple imputation and maximum likelihood, which are readily available in various statistical software packages [4,11]. However, when used on cross-sectional data such methods do not fully utilize information found in the temporal relationships within data records. They also have difficulties in imputing missingness in outputs of dynamic systems that result from combinations of linear and nonlinear effects [12]. In more recent years, more advanced approaches have been introduced to directly tackle the temporal aspects of multivariate imputation. One approach is autoregressive (AR) modelling. Liu and Molenaar [12] introduced vector autoregressive (VAR) models with one-step-ahead predictions, and Parrella, et al. [13] applied spatial-dynamic autoregressive models to impute missing values across a cluster of air pollution monitors. To better capture the complex distributions found in multivariate times series new machine-learning (ML) based approaches, such as generative adversarial networks (GAN) and recurrent neural networks (RNN) show promise [14,15]. Machine-learning methods require only an implicit assumption about the relationships between model variables, treating the system of inputs as a black-box.

In this study, we seek to find the optimal approach for imputing missing values by comparing explicit autoregressive techniques to flexible implicit machine learning techniques. There are many studies that model and analyse household energy consumption and corresponding indoor conditions. Such energy modelling techniques attempt to derive a universal model to describe the thermal properties resulting from building, occupancy and thermal changes over time [16,17]. Few models, however, have been developed with the aim of imputing incomplete high-resolution indoor temperature, humidity or electricity consumption records. As our goal is to maximize imputation performance, our approach focuses on imputing missing values by predicting internal temperatures over time using the information available. The next section describes each of the conventional imputation methods compared with a detailed overview of the proposed spline regression method.

### 3.2. Methodologies Considered for This Study

Ten different models were considered for imputation of missing indoor temperature and humidity datapoints, as listed in Table 2 below. Six conventional imputation methodologies were compared against four variations of the spline regression imputation proposed by this study. The variables used for each model are described in the table below.

#### 3.2.1. Conventional Methodologies

Mean imputation. This method generates a single value, typically by mean or mode, to replace missing temperature data. This can lead to bias and underestimation of standard errors if data are not MCAR. This would give the same mean value as the method of just ignoring missing observations, but standard errors in this assessment would be different owing to the difference in the sample size. This method will be performed to demonstrate the lower limit of imputation performance.

Multiple Imputation of Chained Equations (MICE). This method uses other variables and records from the dataset (between-record) to predict the missing values. It therefore draws on the characteristics of other dwellings (e.g., temperature, energy consumption, building characteristics, demographics, etc.) to predict internal temperatures. This method is now well established and follows a standardised process for imputation. Multiple imputation presents several problems. Firstly, data must be MAR or MCAR. Secondly, in our RCT context where missingness will occur in both the control and intervention groups, using multiple imputation without controlling for group may bias the results or lose the differentiation between groups. However, controlling for the dwelling being part of the control or intervention group would require the analysis to be unblinded before the end of the study-period. This does not follow good clinical research practice and would compromise trial integrity. For these reasons, across record imputation is excluded as an acceptable approach. This study utilized MICE package in R with default settings to impute the missing values for each household with access to external temperature and energy consumption, but not other households.

The primary dataset is a time series with large, chronological missingness; this can also be seen as a forecasting problem. Two state-of-the-art univariate time series methods that detect and model changes in a single variable over time are considered.

Pattern Sequence Forecasting (PSF) is a method proposed by Martínez–Álvarez et al. [18] that identifies periodic patterns in time series data. Bokde, et al. [19] modified PSF to an imputation-friendly version that imputes missing values by taking an average of forecasted and backcasted values. The *imputePSF* package in R was applied to impute missing values one dwelling at a time.

Seasonal Split (SS) is a univariate time series imputation method that splits a time series into seasons and performs imputation for each season. The imputeTS package in R is used to impute the missing values one dwelling at a time, with seasonality set at 48 data intervals per day (daily seasonality).

K-nearest neighbour (KNN) is a widely-used supervised machine learning algorithm where *k* closest neighbours (datapoints) are used to estimate the dependent variable. It is non-parametric, meaning no assumptions are made about the dynamics of underlying data. KNN is more effective when a large dataset is available and is resilient to noisy data, making it suitable for comparison in this study. In this study, nearest observations are based on the closest matching external temperature, energy consumption, and time of day within the same dwelling. *FNN* package in R was used to implement the KNN regression. Parameter *k* is set for each run to be $\sqrt{n}$, where n is the size of training dataset, as recommended by Lall and Sharma [20].

#### 3.2.2. Proposed Approach

Hourly indoor temperature is affected by aspects beyond external temperature and building characteristics, including behavioural and social characteristics of occupants [21,22,23]. A study by Gill, Tierney, Pegg and Allan [23] found that occupant behaviour can account for as much as half of heating energy consumption, emphasizing the need to consider occupant behaviours in modelling internal temperature for this study. In recent years many studies have emerged that model both the physical as well as the socio-behavioural dynamics [24,25,26,27].

There are two distinguishing factors that differentiates our study from the main body of literature. First, while much of the literature in this field attempts to accurately model energy consumption from social and physical determinants, we are inversely attempting to model indoor temperature from physical characteristics of dwelling and environment, energy consumption, and other social determinants. Second is that most studies attempt to create a generalized ‘universal’ model that can describe any given household, but we are creating a model unique to each household. This is due to the difference in the fundamental goal of the study: we hope to best reproduce the half-hourly indoor temperature of each household, and not necessarily understand the underlying dynamics.

A spline regression model would incorporate both the within-variable seasonality and the across-variable relationship between external and internal conditions. Figure 1 shows the structure of relationship between measured variables in our study. External temperature at time *j* (*x_j_*) has a direct and most significant causal relationship to indoor temperature of household *i* (*t_i_*_,*j*_) [28]. This effect of external temperature on internal temperature is moderated by the thermal efficiency and other characteristics of the dwelling *i* (*d_i_*) and energy consumption (*e_i_*_,*j*_), with the presumption that some of the energy will be used to heat the dwelling. Energy consumption is further determined by the occupancy status (*o*_i,j_), which indicates whether a person is present within the dwelling, and by occupant preferences regarding thermal comfort.

A simple linear regression may not sufficiently capture the rich dynamics of this process. Rather, we propose using a spline regression to capture the unique profile of internal temperature that repeats every 24 h for each household. The remaining deviation is captured by the remaining two measured variables, namely energy consumption and occupancy. Spline regression is a non-parametric technique that divides the data into smaller bins at a fixed interval of “knots”, from which a regression model is fitted for each segment [29]. In the case of time series, cubic splines with constraints on continuity of first and second derivatives can sufficiently model the smooth but erratic nature of change, such as those found in half-hourly internal temperature [30]. We model a cubic polynomial g for internal temperature defined on endpoints $\left[h_{0},h_{24}\right]$ with regards to knots ${\left\{\xi_{i}\right\}}_{i=1}^{m}$ where:(1)$$g\left(x\right)=d_{i}x^{3}+c_{i}x^{2}+b_{i}x+a_{i},\;x\in \left[\xi_{i}\;,\;\xi_{i+1}\;\right]$$
where $i=0:m,\;\xi_{0}=h_{0}\;\mathrm{and}\;\xi_{m+1}=h_{24}\;$ and x is time in hours of each household. Cubic spline has two key advantages over conventional polynomial regression. First, outliers can significantly skew the results of polynomial regression globally, while for spline regression, the effect is locally contained to the corresponding spline. In the case of cubic spline, the additional constraints of continuity and continuity in the first and second derivatives minimize the chance of “wriggle” behaviour commonly seen in overfitted models.

In this study, three additional models with successively added independent variables are introduced to observe their explanatory power, as well as a simple linear regression model for comparison.

Baseline linear regression model (R0). As a baseline regression approach, we model the primary effect of external temperature on indoor temperature as a simple linear model:(2)$$Tint_{i,j}=\beta_{0,i\;}+\beta_{1,i\;}x_{e,j}+\varepsilon_{i}$$

As demonstrated in Figure 2, there is a positive correlation between internal and external temperature. Each datapoint represents a 30-min internal temperature reading for one dwelling with all datapoints over a 90-day period shown. While a linear regression fit captures the correlation between the two, it fails to capture the full range of variance. This is particularly evident in household 3 of Figure 2, where there are two clusters of indoor/outdoor temperature relations.

Baseline spline regression model (SR1). The second baseline model attempts to better capture the dynamics of internal temperature throughout the day. This is done by modelling the internal temperature profile of a dwelling over time, in our case at every half-hour, as shown in Figure 3. We set the number of knots m to be 11, where we fit a cubic polynomial for each 2 h intervals of data to best capture the changes in internal temperature for each household, as shown in Figure 3.

Improved spline regression models (SR2,3,4). We iteratively test the marginal improvements of including additional independent variables to the baseline spline model. Half hourly external temperature and half hourly energy consumption measured for each household are iteratively included as independent variables. A third dummy variable, occupancy status, is also tested for interaction with energy consumption. This variable is created by categorizing the time series energy consumption data into two status groups (present/absent) for each dwelling using cluster analysis. A total of three variations in model are created for testing, as summarized in Table 2.

## 4. Data

In order to measure the performance of different imputation methodologies, we utilized the dataset of 99 households from the 2018 study year, when there were no issues with logger batteries and all dwellings had complete temperature and humidity data. As this dataset does not have any missingness, we created test datasets by artificially deleting values in similar fashion to the missingness characteristics of the 2018 data. Creating these datasets allows us to test the imputation performance of each method against actual values. In this section, we discuss the method of selecting datapoints for deletion with the aim of replicating missing data from field trials.

The time at which a battery fails was approximated by a Weibull distribution [31], capturing the increased likelihood of battery failure over time using an approximate hazard function. We also accounted for the fact that the variation in mean time to failure (MTTF) has an upper limit bounded by the physical specifications of the battery. Given these two characteristics, we propose a Weibull function with negative skewness (*k* > 3.7) to describe the probability of failure over time:(3)$$f\left(t\right)=\frac{k}{\lambda }{\left(\frac{t}{\lambda }\right)}^{k-1}e^{-{\left(\frac{t}{\lambda }\right)}^{k}}$$
where k is the shape parameter and λ is a scale parameter. The scale parameter is the point where 63.2% percent of the population will have failed, regardless of the shape parameter. The λ is estimated to be 76 days from the 2019 records that had missing data, out of the total 92 days. λ is set to be 30 days for the simulated deletion points and k is set to be 6 (slight negative skewness), which would retain the shape of distribution but shift it to the left. The resulting distribution of failure time can be seen in Figure 4. This would start the missingness earlier, allowing creation of a wider range of missingness levels to test with. We also assumed that after a certain period, all of the batteries are replaced at the same time but prior to the end of the study period. This allows for a process of batteries failing at different times with increasing likelihood of failure as time progresses. It is assumed that all dataloggers within the test dataset fail and batteries are replaced before the end of the study period.

The timing of battery replacement was used to determine the percentage of total missingness within the sample, with the assumption that all batteries are replaced on the same day. Four different test datasets were generated at varying levels of missingness (10%, 30%, 50% and 70%). Figure 5 shows the resulting internal temperature data with missingness for one household.

## 5. Results

The eleven models identified in Table 2 are applied to four datasets, resulting in 44 models to compare. The performance of each run is calculated based on the normalized mean absolute error (NMAE) and root mean squared error (RMSE):(4)$${\mathrm{NMAE}}_{i}=\frac{1}{n_{i}}\;\overset{n_{i}}{\underset{j=1}{\sum }}\frac{\left|t_{i,j}^{ac}-t_{i,j}^{imp}\right|}{V_{i}^{max}-V_{i}^{min}}$$
(5)$${\mathrm{RMSE}}_{i}=\sqrt{\overset{n_{i}}{\underset{j=1}{\sum }}\frac{{\left(t_{i,j}^{ac}-t_{i,j}^{imp}\right)}^{2}}{n}}$$
where $n_{i}$ is the number of imputed data points for dwelling *i*, $t_{i,j}^{ac}$ is the actual internal temperature of household *i* at time *j*, $t_{i,j}^{imp}$ is the imputed internal temperature of dwelling *i* at time *j*. $V_{i}^{max}$ and $V_{i}^{min}$ are maximum and minimum values for the range of actual recorded internal temperatures for the given dwelling. Figure 6, Figure 7 and Figure 8 show the actual vs. imputed values for a single dwelling for each of the different models used.

### 5.1. Imputation Characteristics

Figure 6 shows the first three days’ worth of imputation using MI and MICE for Household (1). It can be seen that mean imputation cannot capture variations over time, while MICE generates perturbations within sequential datapoints that should not exist in continued time series measurement of indoor temperatures. This is due to MICE’s over-emphasis on using between-record characteristics while ignoring within record characteristics. The added noise makes MICE unsuitable for imputing missing time series where change over time is of interest.

Figure 7 shows the results from the two univariate time series approaches. It can be noted that PSF is unable to capture the consistent structure of daily seasonality. The seasonal split (SS) approach does identify the daily seasonality; however, the daily imputations are identical over the three-day period and beyond.

Figure 8 shows the imputation results from the spline regression approaches. The linear regression (R0) modelling internal temperature with external temperature manages to capture the cyclical nature of indoor temperature but fails to cover the minimum and maximum range. In the baseline spline regression model (SR1) that models internal temperature as a cubic spline over the period of 24 h, the daily pattern is efficiently covered, however as this approach does not take into consideration any other causal factors, it cannot model any long-term changes over time such as change in season.

Figure 9 shows the imputation results from KNN and spline regression SR3. Both methods show marked improvement compared to previous approaches shown, and show similar visual performance. It should be noted that both KNN and SR3 have same level of access to datasets (external temperature, energy consumption and internal temperature).

### 5.2. Imputation Performance

Individual models were developed for each of the 99 dwellings for all methods tested. Table 3 and Figure 10 show the mean of RMSE from the 99 dwellings tested for the temperature data. As expected, RMSE increases as levels of missingness increase and the amount of information available to impute decreases.

A notable improvement in imputation performance occurs in the spline regression approach when external temperature is included (SR2), resulting in a 6% reduced RMSE compared to the baseline model (SR1). Additional improvements are also observed when energy consumption (SR3) and occupancy status (SR4) is included in succession, with 7% reduced RMSE and 8.6% reduced RMSE compared to the baseline model (SR1), respectively.

It can be seen that spline regression models the dynamics of internal temperature over time for each household better than other models at all missingness levels. At 10% missingness, the best spline regression model (SR4) has on average 14% and 16% smaller RMSE compared to MICE and univariate time series (SS) methods, respectively. While this gap remains consistent for MICE, the gap between SR4 and SS widens to 22% smaller RMSE when more than half of data is missing.

The gap in performance between the next best performing method (KNN) and spline regression widens as levels of missingness increases, from 1.5% to 4.5% smaller RMSE for 10% and 70% missingness, respectively.

Similar results can be observed from the comparison of NMAE results, as shown in Figure 11; across all missingness levels, spline regression models perform better than the conventional models tested for comparison. It should be noted that due to the nature of NMAE’s dependency on maximum value $V_{i}^{max}$ and minimum value $V_{i}^{min}$ for each sample data range, comparison across missingness levels is inappropriate. The best performing spline regression (SR4) outperforms MICE with ~16% smaller NMAE at all missingness levels, while outperforming SS with 19% to 24% smaller NMAE across the missingness levels. SR3, with the same level of access to datasets as KNN, is negligibly outperformed by KNN (0.6% larger NMAE) at 10% missingness level. As missingness levels increase, SR3 overtakes KNN with 2% smaller NMAE at 70% missingness. Detailed results can be seen in Table 4.

## 6. Discussion and Conclusions

We explored the different imputation methods for addressing large, chronologically missing time series datasets in a randomized controlled trial. A set of spline regression models were used to capture the dynamics of high-resolution data on internal temperature, electricity consumption and outdoor temperature. Results showed that all of the spline regression models perform better than conventional methods at all tested levels of missingness. Spline regression models performed best when all causal factors available were included in the model, i.e., the time of day, external temperature over time and electricity consumption over time, as well as modelled occupancy. The interaction between modelled occupancy and electricity consumption (SR4) was found to marginally improve the results, however, in a real-world application this interaction factor could be removed to simplify the model without a large effect on results. For cases where uninterrupted missingness is short (<10% of data), results suggest that KNN can also be used without significant performance loss as compared to the proposed Spline regression approach.

Another issue to note is that from the within-record perspective, the univariate time series seasonal split (SS) method performs significantly better in capturing the inter-day temperatures compared to other conventional methods. However, as SS does not take into consideration the external environmental conditions for the imputed period, its projection may be error-prone if the external temperature conditions change during the imputed period. While our RCT is performed only during a relatively stable winter period, this may be problematic for studies where climate conditions change over different seasons.

It should also be noted that this paper investigates imputation strategies specifically for a long, uninterrupted missingness, found in situations where device failures cannot be identified easily. For the more common missingness that are short and frequent (e.g., network issues that drop small pockets of data at random intervals), time-series modelling that combines forecasting and back-casting such as PSF might be more suitable. Researchers utilizing spatial environmental data may also find useful parallels for their imputation efforts, in terms of capturing continuous but nonlinear patterns in both temporal and spatial data [32,33].

While there is no method that can perfectly model or non-parametrically impute the internal temperature, this study has shown how understanding and utilization of the available dataset’s across-record, across-variable and within-variable dynamics can improve the imputation performance. Given the findings, the spline regression approach was shown to be the best method at imputing missing temperature readings.

Future work should investigate methods to better capture the within-record seasonality up to the standards of seasonal split in regression models. Methods in transformation of other independent variables that would best capture the relationship between variables should also be explored, as we have done with converting time series into repeated daily seasonality.

## Figures and Tables

**Figure 1 ijerph-19-01307-f001:**
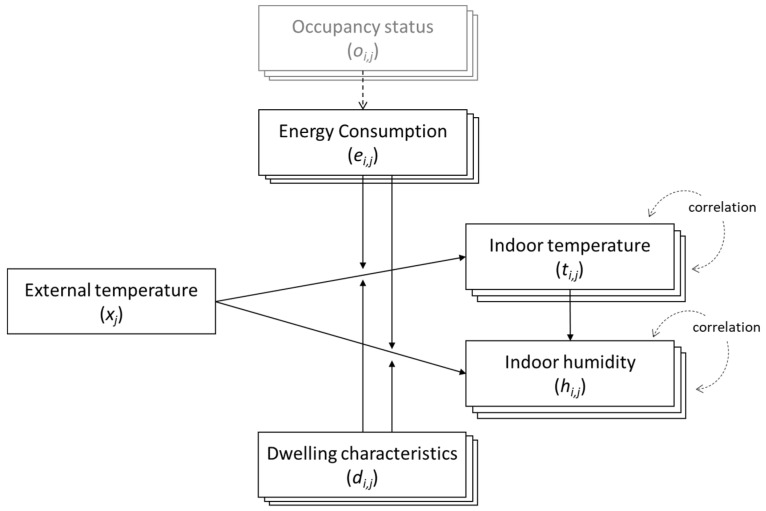
Relational dynamics between variables collected for dwelling *i* at time *j*.

**Figure 2 ijerph-19-01307-f002:**
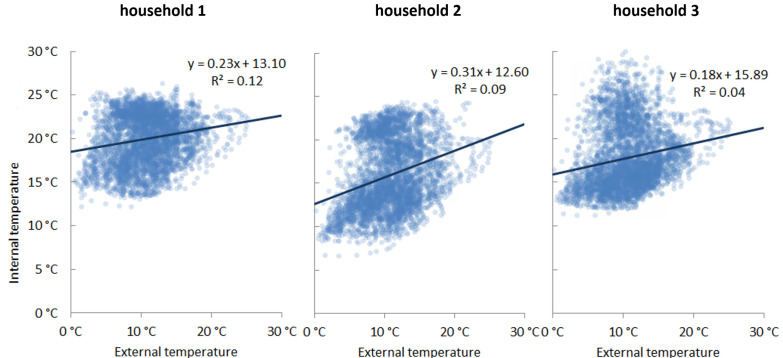
Relationship between external vs. internal temperatures from three sample dwellings.

**Figure 3 ijerph-19-01307-f003:**
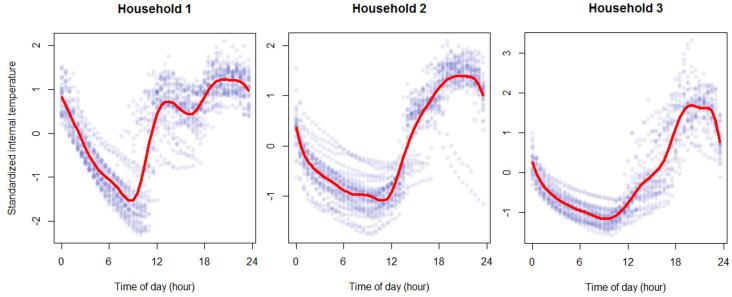
Relationship between the time of day vs. internal temperature from three sample dwellings, with cubic spline fit (m = 11).

**Figure 4 ijerph-19-01307-f004:**
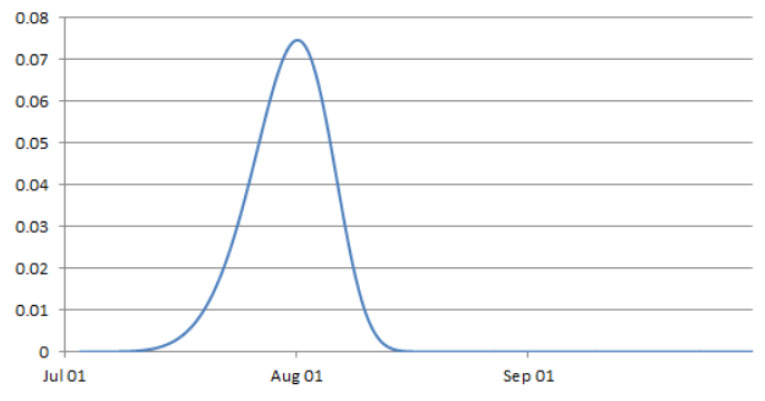
Probability density function of modelled battery failure rate over time (*k* = 6, *λ* = 30 days).

**Figure 5 ijerph-19-01307-f005:**
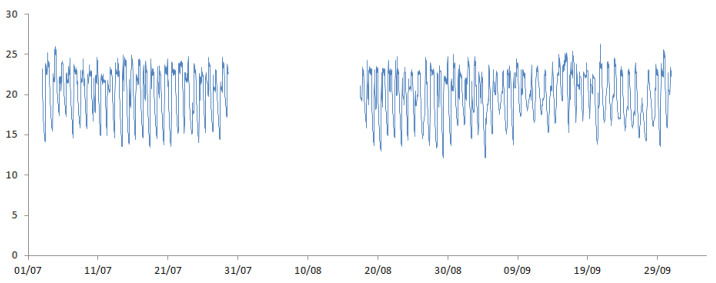
Recorded internal temperature of 1st household with generated missingness.

**Figure 6 ijerph-19-01307-f006:**
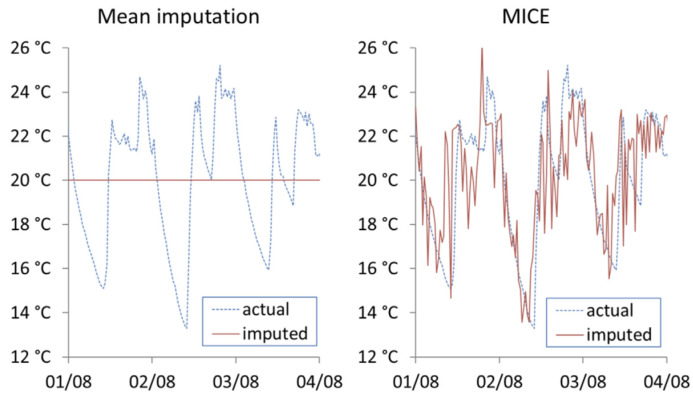
Actual vs. imputed internal temperature from mean imputation and MICE.

**Figure 7 ijerph-19-01307-f007:**
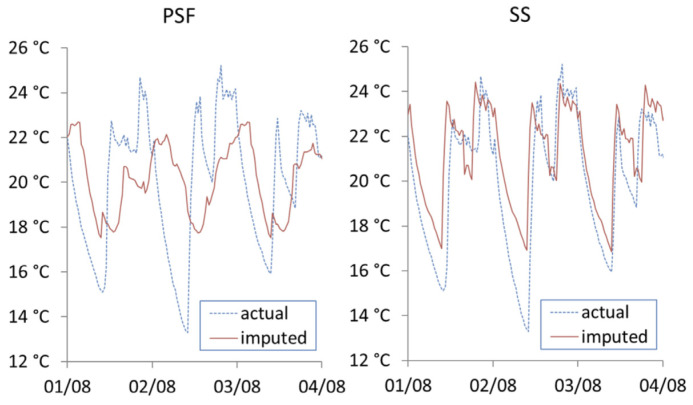
Actual vs. imputed internal temperature from univariate time series imputation methods.

**Figure 8 ijerph-19-01307-f008:**
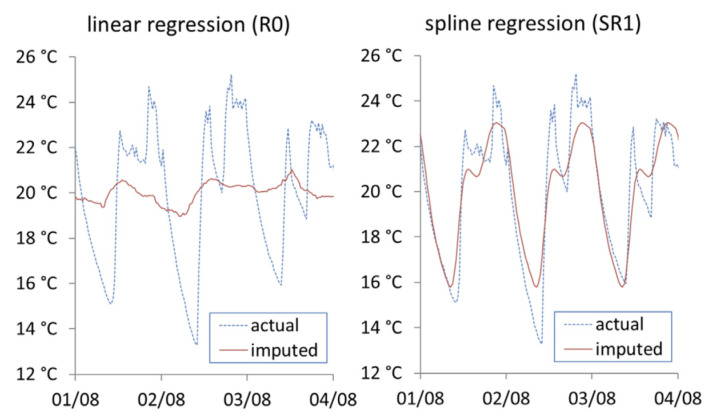
Actual vs. imputed internal temperature from linear regression (R0) and the baseline spline regression (SR1).

**Figure 9 ijerph-19-01307-f009:**
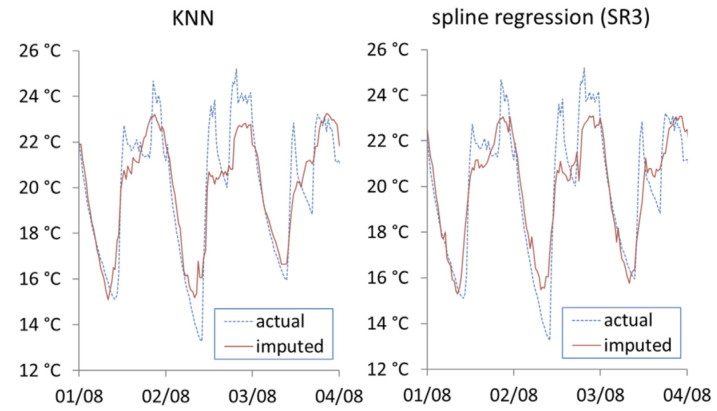
Actual vs. imputed internal temperature from KNN and spline regression (SR3).

**Figure 10 ijerph-19-01307-f010:**
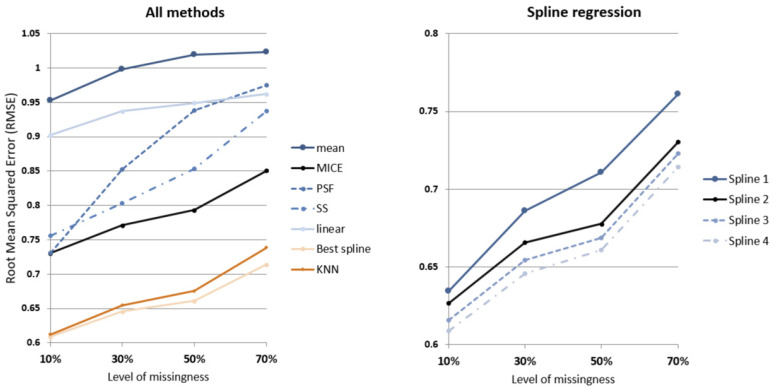
Imputation performance (mean RMSE) of tested methods by levels of missingness.

**Figure 11 ijerph-19-01307-f011:**
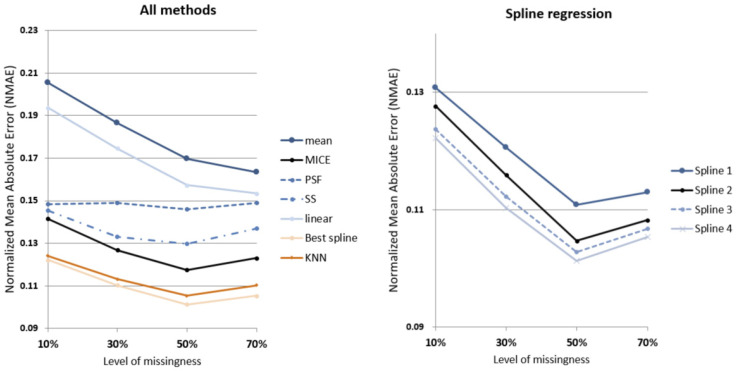
Imputation performance (NMAE) of tested methods by levels of missingness.

**Table 1 ijerph-19-01307-t001:** Summary of data available for imputation.

Variable	Description	Source
External temperature	External temperature *x_j_* at time *j*, in °C	The Australia Bureau of Meteorology
Internal temperature	Internal temperature *t_i,j_* household *i* at time *j*, in °C	Data logger for household *i*
Indoor humidity	Internal humidity *h_i,j_* of household *i* at time *j*, in %	
Electricity consumption	Electricity consumption *e_i,j_* of household *i* at time *j*, in kWh	Energy utility for household *i*

**Table 2 ijerph-19-01307-t002:** List of imputation methodologies considered for this study.

Methodology			Variables Used				
Type	Name		T_int_	T_ext_	Energy	Time	Occupancy
Baseline	Mean imputation	(MI)	✓	-	-	-	-
Multivariate	Multiple imputation by chained equations	(MICE)	✓	✓	✓	-	✓
Univariate time series	Pattern sequence forecasting	(PSF)	✓	-	-	-	-
	Seasonal split	(SS)	✓	-	-	-	-
Machine learning	k-nearest neighbour	(KNN)	✓	✓	✓	✓	-
Linear regression	t ^a^~x ^b^	(R0)	✓	✓	-	-	-
Spline regression	t~sp(hour ^c^) ^d^	(SR1)	✓	-	-	✓	-
	t~sp(hour) + x	(SR2)	✓	✓	-	✓	-
	t~sp(hour) + x + energy ^e^	(SR3)	✓	✓	✓	✓	-
	t~sp(hour) + x + (energy) x (occupancy ^f^)	(SR4)	✓	✓	✓	✓	✓

^a^ t—internal temperature in degree Celsius. ^b^ x—external temperature in degree Celsius. ^c^ hour—time of day in hour, from 0 to 24. ^d^ sp—cubic spline function. ^e^ Energy–electricity used every half hour, in Watts. ^f^ Occupancy—dummy variable (present/absent) based on energy use. Note: all datapoints are in half-hour intervals.

**Table 3 ijerph-19-01307-t003:** Mean RMSE of imputation methods by missingness level for temperature data.

Methods		Missingness Level			
		10%	30%	50%	70%
Baseline	Mean Imputation	0.9529	0.9981	1.0190	1.0231
Multivariate	MICE	0.7303	0.7712	0.7934	0.8506
Univariate time series	PSF	0.7310	0.8519	0.9381	0.9376
	Seasonal split	0.7554	0.8030	0.8536	0.9376
Machine learning	K-nearest neighbours	0.6121	0.6546	0.6754	0.7387
Linear regression	t ^a^~x ^b^	0.9021	0.9373	0.9493	0.9624
Spline regression	t~sp(hour ^c^) ^d^	0.6346	0.6860	0.7109	0.7609
	t~sp(hour) + x	0.6268	0.6657	0.6777	0.7304
	t~sp(hour) + x + energy ^e^	0.6160	0.6545	0.6687	0.7228
	t~sp(hour) + x + (energy) x (occupancy ^f^)	0.6089	0.6458	0.6608	0.7143

^a^ t—internal temperature in degree Celsius. ^b^ x—external temperature in degree Celsius. ^c^ hour—time of day in hour, from 0 to 24. ^d^ sp—cubic spline function. ^e^ Energy—electricity used every half hour, in Watts. ^f^ Occupancy—dummy variable (present/absent) based on energy use. Note: all datapoints are in half-hour intervals.

**Table 4 ijerph-19-01307-t004:** Mean NMAE of imputation methods by missingness level for temperature data.

Methods		Missingness Level			
		10%	30%	50%	70%
Baseline	Mean Imputation	0.2056	0.1864	0.1697	0.1634
Multivariate	MICE	0.1416	0.1269	0.1175	0.1230
Univariate time series	PSF	0.1483	0.1490	0.1460	0.1491
	Seasonal split	0.1454	0.1332	0.1298	0.1369
Machine learning	K-nearest neighbours	0.1241	0.1131	0.1054	0.1103
Linear regression	t ^a^~x ^b^	0.1937	0.1746	0.1574	0.1535
Spline regression	t~sp(hour ^c^) ^d^	0.1308	0.1207	0.1109	0.1130
	t~sp(hour) + x	0.1277	0.1159	0.1047	0.1083
	t~sp(hour) + x + energy ^e^	0.1238	0.1123	0.1028	0.1068
	t~sp(hour) + x + (energy) x (occupancy ^f^)	0.1223	0.1103	0.1013	0.1054

^a^ t—internal temperature in degree Celsius. ^b^ x—external temperature in degree Celsius. ^c^ hour—time of day in hour, from 0 to 24. ^d^ sp—cubic spline function. ^e^ Energy—electricity used every half hour, in Watts. ^f^ Occupancy—dummy variable (present/absent) based on energy use. Note: all datapoints are in half-hour intervals.

## Data Availability

Data is not publicly available due to privacy requirements of the study.

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
