# Peer review of "Strategies for Imputation of High-Resolution Environmental Data in Clinical Randomized Controlled Trials"

_ijerph, 2022, doi:10.3390/ijerph19031307_

Round 1
Reviewer 1 Report
This is a well-done piece of research on a niche, but important, topic. As the volume of spatial and temporal environmental data collected via sensors increases rapidly, missing data becomes a key challenge for analysis. This paper makes a nice scientific contribution by testing and reporting on various approaches to incorporating time series profiles into imputation of time-series environmental data. The paper is nicely organized, well-written, and contextualized in the literature (though see my comments below). The methods appear sound and the results are of interest.
Some comments:
- Some information about the study setting would be useful, particularly given the modeling is of temperature. What country? What kind of climate are these data taken from? Urban, suburban, rural?
- I’m a bit confused by the KNN. What are the nearest neighbors to an observation – observations at the same dwelling but ‘nearby’ in time? How is this parameterized? More explanation and details would be useful.
- The authors say missing data may extend for several days to several months in a single dwelling. It would seem different approaches would be best for different types of scenarios. Data missing for shorter times would probably be easily modeled using within-dwelling data – e.g. the nearest observation or simply a moving average, and by incorporating the diurnal cycle. For longer time periods, external data, such as outdoor temperature may be more informative. This deserves some discussion.
- I am not surprised that knowing the outdoor temp, energy consumption, occupancy, etc. is useful for imputing indoor temperature. But having these kinds of data external to a research project may be unlikely in many research scenarios. Although the spline regression outperforms k-means when including these covariates, it appears that its performance over k-means is marginal. The takeaway for me reading this was that k-means performs quite well, with few assumptions or external data requirements. This might be useful for readers.
- Spline often has best fit in such comparisons of estimations because it ‘rubber sheets’ a curve that passes directly through known observations. It appears tests were done using different levels of missingness, addressing this issue to some extent, but is over-fitting a concern?
- There are some useful analogies of this research to research on the imputation of spatial environmental data. The paper would be improved by briefly incorporating these into the literature review and perhaps in the discussion as a point of comparison – see, for example, Mennis et al. (2018), appearing in this journal, and work by Henry and Boscoe (2008) appearing in International Journal of Health Geographics, among others.
Reviewer 2 Report
Minors
292 - I believe it is t_{e,j}
312 - What is the c_i in the model 1?
411 - "Error! Reference source not found."
Majors
- The variables names in figure 1 and model 2 are different. It should be standardized.
- I was a little confused by the names of the covariates. Are "T_{ext}" and "ext" (table 3 and 4) the same variables? What's the difference between them?
- Was the regression model fitted with any covariates? Would that improve your performance?
- Is NMAE metric a good choice for this context? It can be seen in Figure 10 that it decreases as the level of missingness increases (up to the 50% level), which does not seem intuitive. What is the explanation for this?
- To create artificial data similar to 2019, using 2018 data, the weibull distribution was used. Is it possible to see how this distribution fits the 2019 data?
Round 2
Reviewer 2 Report
I was completely satisfied with the authors' answers.